# Analyzing Cultural Routes and Their Role in Advancing Cultural Heritage Management within Tourism: A Systematic Review with a Focus on the Integration of Digital Technologies

Eleftheria Iakovaki [1], Markos Konstantakis [2,*], Alexandros Teneketzis [3] and George Konstantakis [4]

1 Department of Conservation of Antiquities and Works of Art, University of West Attica, 12243 Athens, Greece; cons20676094@uniwa.gr
2 Department of Cultural Technology and Communication, University of the Aegean, University Hill, 81100 Mytilene, Greece
3 Department of History and Archaeology, University of Patras, University Hill, 26504 Patra, Greece; ateneketzis@upatras.gr
4 Department of Natural Sciences, Hellenic Open University, 26335 Patra, Greece; std072784@ac.eap.gr
* Correspondence: mkonstadakis@aegean.gr

**Abstract:** This review constitutes a comprehensive systematic review analyzing cultural routes, with a particular focus on the concept of the cultural route as a tourist–cultural product. Within this framework, the paper offers an overview of contemporary technological challenges, concerns, and limitations. It thoroughly explores cutting-edge technologies pertaining to the promotion of cultural heritage, both in general and in the specific context of realizing the concept of the cultural route, a tourist–cultural service enriched by the utilization of new media. Additionally, it extensively references the latest techniques and models for enhancing the user experience of digital cultural tourism products. Moreover, the paper showcases existing digital platforms and tools that encapsulate and emphasize the notion of cultural tourism. It assesses the respective methodologies, technologies, and techniques employed in each case, accompanied by illustrative instances of their applications. Finally, an empirical evaluation was conducted focusing on user needs and expectations during a cultural route.

**Keywords:** cultural routes; cultural heritage; cultural management; analysis; digital culture; tourism

## 1. Introduction

Contemporary trends within the realms of culture and tourism underscore the growing desire among travelers to venture beyond conventional experiences. Simultaneously, advancements in information and communication technologies have facilitated easier access to cultural wealth, shaping the demand for fresh cultural offerings. The role of cultural routes in this context is undeniably crucial, necessitating their utilization by cultural institutions and their integration with the tourism sector.

This review constitutes a comprehensive systematic review within the field of cultural tourism, with a particular focus on the concept of the cultural route as a tourist–cultural product. It scrutinizes the cultural route through the lens of cultural tourism, underscoring the cultural heritage of a location as a pivotal element in crafting a tourist–cultural itinerary. Furthermore, the paper delves into the emergent trends within cultural tourism and the dynamic interplay between the culture and tourism sectors, adopting a comprehensive and pervasive perspective. Concurrently, it identifies and categorizes the pertinent practical efforts within both the academic literature and the cultural and tourism industries.

Within this framework, the paper offers an overview of contemporary technological challenges, concerns, and limitations. It thoroughly explores cutting-edge technologies pertaining to the management of cultural heritage, both in general and in the specific context of realizing the concept of the cultural route, a tourist–cultural service enriched by the

utilization of new media. Additionally, it extensively references the latest techniques and models for enhancing the user experience of digital cultural tourism products. Moreover, the paper showcases existing digital platforms and tools that encapsulate and emphasize the notion of cultural tourism. It assesses the respective methodologies, technologies, and techniques employed in each case, accompanied by illustrative instances of their applications. The main questions that will be analyzed are the following:

- What is cultural tourism, and how does it relate to cultural heritage?
- What are cultural routes, how are they designed, and what information technology tools are used?
- What kind of experiences does the traveler gain when using a cultural route? How is the user's cultural experience defined?
- How do cultural routes advance cultural heritage management?
- What applications and research studies are associated with cultural routes?

Regarding the rest of this review, Section 2 provides an overview of the current landscape in cultural heritage fields. Section 3 discusses the systematic review of cultural routes by addressing PRISMA methodology along with a critical analysis of the applications, while Section 4 presents the results of the research regarding user needs and expectations during a cultural route. In Sections 5 and 6, conclusions and suggestions for future research are presented.

## 2. Background Knowledge

### 2.1. Cultural Tourism

The intricate nature of culture and the interconnections between "tourism" and "culture" have contributed to substantial ambiguity in the current body of literature concerning a precise and universally acknowledged definition of cultural tourism [1]. Cultural attractions, whether they manifest as traditional landmarks (e.g., museums, architecture, heritage sites) or as events (e.g., festivals, folklore), are widely regarded as significant catalysts for traveler interest and inspiration. Cultural tourism, in theory, is regarded as a long-standing tourist practice with its roots in the 16th-century British Grand Tour [2]. Defining cultural tourism proves challenging due to the complexity of its elements and characteristics. Several early authors [3,4] view cultural tourism as a form of specialized interest travel, where tourists seek new and authentic experiences, including aesthetic, intellectual, emotional, or psychological dimensions. According to Adams [5], who provides a broad definition, cultural tourism is seen as a form of travel for personal enrichment.

Focusing on definitions originating from the academic world, a clear-cut definition of cultural tourism as "the movement of people for strictly cultural reasons" can be established. However, consensus is elusive when attempting to provide a precise interpretation of cultural motives. This lack of alignment among different definitions stems from the ambiguity inherent in the concept of cultural motivation. Richards [6] notes that despite its historical association with cultural heritage in Europe, cultural tourism now increasingly includes folk culture and other intangible forms of heritage [7–9]. Furthermore, Richards and Wilson [10] argue that not only intangible heritage but also other cultural products, such as virtual locations or structures, thematic attractions, or large-scale events, have become fully integrated into this definition. Kim et al. [11] even include festivals and musical attractions, amusement parks, local festivals, exhibitions, and aesthetic/cognitive attractions within the scope of cultural tourism.

In summary, culture is a dynamic concept that varies across time and space, thus lending itself to various parameters. While some general agreement exists regarding the definition of cultural tourism as tourism driven by cultural consumption, questions and differences in criteria arise when pinpointing cultural consumption practices (i.e., consumption of cultural heritage, engagement in cultural activities, and cultural experiences). This complexity adds nuance to the definition of cultural tourism.

*2.2. Cultural Routes*

Therefore, with defined and presented the basic characteristics of cultural tourism, we will discuss in this section one of the most important tools for the management, promotion, and interpretation of the cultural heritage of a place, the highlighting of common cultural elements of different places, the encouragement of collaborations, intercultural dialogue, and sustainable development: cultural routes.

Based on recent case studies illustrating the different aspects of the relationship between tourism and cultural heritage [12,13], cultural routes offer an experiential means of comprehending, by means of personal investigation and direct experience, the aspects that simultaneously distinguish and unite us, therefore creating an environment conducive to innovation and speculation. Cultural routes not only connect certain regions but also unite different cultures. In the past, they were the main routes for the transfer of goods and ideas, elements of intangible and material heritage worldwide.

A "cultural route" refers to a collection of Points of Interest (POI) that encompass a particular geographic area and share comparable attributes, a central theme, and distinctive architectural or historical features in addition to facilities, natural landscapes, and buildings. In contemporary times, an enhanced iteration of the cultural route endeavors to establish a link between the past and the present of a given region, city, or settlement.

According to Meyer [14], cultural routes can attract new visitors as well as repeat ones. The dispersal of visitors and the distribution of income from tourism result in a more equitable distribution of economic benefits, and increasing the length of stay and tourist spending generates greater profit. Furthermore, the connection of many attractions that individually may not attract visitors is a key advantage of a cultural route. The synergies created promise greater attractiveness and lead to various benefits, such as increasing the overall appeal of a destination and enhancing the sustainability of a tourism product. Managing transportation capacity is facilitated due to the dispersion of tourists, and negative environmental impacts are reduced as a result of this dispersal.

The content of a cultural route can vary depending on the characteristics of the region and the extent of its cultural and natural heritage. More specifically, from the literature review, it is evident that cultural routes can be categorized thematically into thematic routes (routes that focus on a specific thematic element), historical routes (routes that revolve around monuments and cultural elements of a specific time period or era), and mixed cultural routes (routes that include elements of both cultural and natural heritage, regardless of type or era, as part of the overall cultural landscape of a place). In addition, depending on their scale and extent, cultural routes can be classified as urban routes (involving only the monuments and other elements of an urban area), local routes (operating within a limited geographical unit, such as within the boundaries of a municipality), sub-regional routes (extending over a relatively broader geographical area, mainly at the regional and inter-regional levels), national routes and cross-border routes [15,16].

Within this particular framework, the notion of a cultural route emerges as intricate and multifaceted. It serves as an initial qualitative contribution to the notion of cultural heritage preservation, encompassing both natural and cultural components. On a theoretical level, a cultural route can be conceptualized along temporal and spatial axes, functioning as a geographical manifestation of continuity predicated on the dynamics of movement or the concept of exchange. Conversely, the tangible dimensions of a cultural route manifest a different story. Therefore, the placement in the territorial network is the main force of cultural routes. The route is divided into sub-networks and site networks that share common cohesion. The search for this common cohesion and continuity is very important regarding the image and promotion of the destination. Cultural tourism is not limited to a series of visits to places.

Additionally, developments in the fields of information and communication over recent decades offer easier access to cultural heritage. However, they also diversify the preferences of modern travelers, shaping the demand for new cultural goods and educational forms of entertainment (edutainment).

*2.3. Cultural User Experience*

User Experience (UX) is related to the overall functionality and usability of an interactive digital application. In recent years, user experience has been recognized as a critical factor for the acceptance not only of interactive systems but also of any industrial product and service. In the context of developing interactive digital applications, the term covers all aspects of a user's experience with a computer system, such as graphic design, user interface, physical interaction, and user manuals. Undoubtedly, user experience research can be characterized as very complex, involving many different parameters, such as usability, credibility, findability, desirability, and more. It is apparent that user experience is a comprehensive notion that extends beyond usability and accessibility to encompass additional dimensions of user acceptance of an interactive system, such as the emotional factor. Cultural User Experience (CUX) is defined as "the unique knowledge and experience generated by different cultural identities" and is influenced by the interplay between various cultural elements and the diverse cultural backgrounds of users [17].

Cultural experience is primarily an educational experience [18]. Museums are open spaces where one can learn regardless of one's age and educational, social, or economic status. In other words, cultural spaces express the ideal of open education because education is the right of all people, and they can enjoy it throughout their lives without any restrictions and according to their personal choices. Therefore, these spaces, as lifelong education institutions, can be considered expressions of learning based on free choice.

Many cultural navigation applications have been designed and evaluated with the aim of improving the user experience. The Svevo Tour application [19] uses storytelling techniques to attract visitors and enhance their cultural experience while touring a literary museum focusing on the famous Italian author Italo Svevo, who lived in Trieste between the 19th and 20th centuries. During the evaluation process, this application examines the relationship between AR technology and storytelling and the appropriateness of using augmented reality in literary museums with adults who have little experience with AR technologies.

Archeoguide, an early application in the cultural sector, provides customized excursions to archaeological sites. It accomplishes this by enhancing information, reconstructing ruined areas, simulating ancient life, and utilizing mobile computing, 3D visualization, and augmented reality techniques [20]. Additionally, Sylaiou [21] explores the behavior of virtual characters in virtual museums (VM) environments. Three user models (curator, guard, and visitor) are used as narrators who introduce participants to an emotional story behind a historical sculpture to enhance the user's cultural experience through personalized interaction [22].

Finally, on the MuseumEye platform [23], information augmentation is based on a hybrid system that combines Simultaneous Localization and Mapping (SLAM) technology and indoor or low-energy Bluetooth (BLE) beacons. These technologies include a combination of multimedia and different levels of visual information required for visitors to navigate the museum. Using mobile devices for the pilot application of the platform, a UX design model was developed to assess the user's experience and usability of the application. User guidance is available through the MuseumEye platform at predefined points in the space via access to multimedia material.

Therefore, user experience is a complex concept, which makes its evaluation complex as well, as it is directly related to the physical interaction of users with an interactive system, product, or service. In recent years, the rapid development of interactive technologies, especially touch-based portable devices, combined with the subsequent emergence of online services and applications for various aspects of human activity, including social networks and augmented reality, as well as the continuous improvement of these technologies and applications in terms of usability and accessibility, have contributed to the emergence of more interaction aspects as significant for evaluating the user experience.

## 3. Systematic Review in Cultural Routes

*3.1. PRISMA Methodology*

Applications, software, and platforms available on the Internet related to cultural routes can find application in various domains, including education and cultural spaces, as well as in the fields of advertising, product promotion, and presentations in professional and academic settings. Certified cultural routes by the Council of Europe's Cultural Routes Program number approximately 33 and are characterized by thematic diversity and geographical dispersion. Among these are the Routes of Santiago de Compostela Pilgrims (1987), the Hanseatic Towns (1991), the Viking Routes (1993), the Phoenician Route (2003), the European Route of Jewish Heritage (2004), the Olive Tree Routes (2005), the Wine Routes (2009), and the European Megalithic Culture Route (2013), among others [24,25].

Therefore, a systematic literature review methodology was employed to assess the current state of the topic, pinpoint research gaps, and propose new research directions. To identify relevant studies, a search was conducted on the Scopus and Google Scholar databases in September 2023, employing specific keywords related to cultural tourism, cultural routes, cultural heritage, and digital applications.

The outcomes of this search are presented in a PRISMA flow diagram [26], as illustrated in Figure 1 below. In adherence to the methodologies outlined by Liberati et al. [26] and Shamseer et al. [27], the present study offers a detailed "explanation and elaboration" of the reporting checklist items from the PRISMA guidelines, which are adhered to in the execution of systematic reviews within the cultural tourism and cultures routes domain.

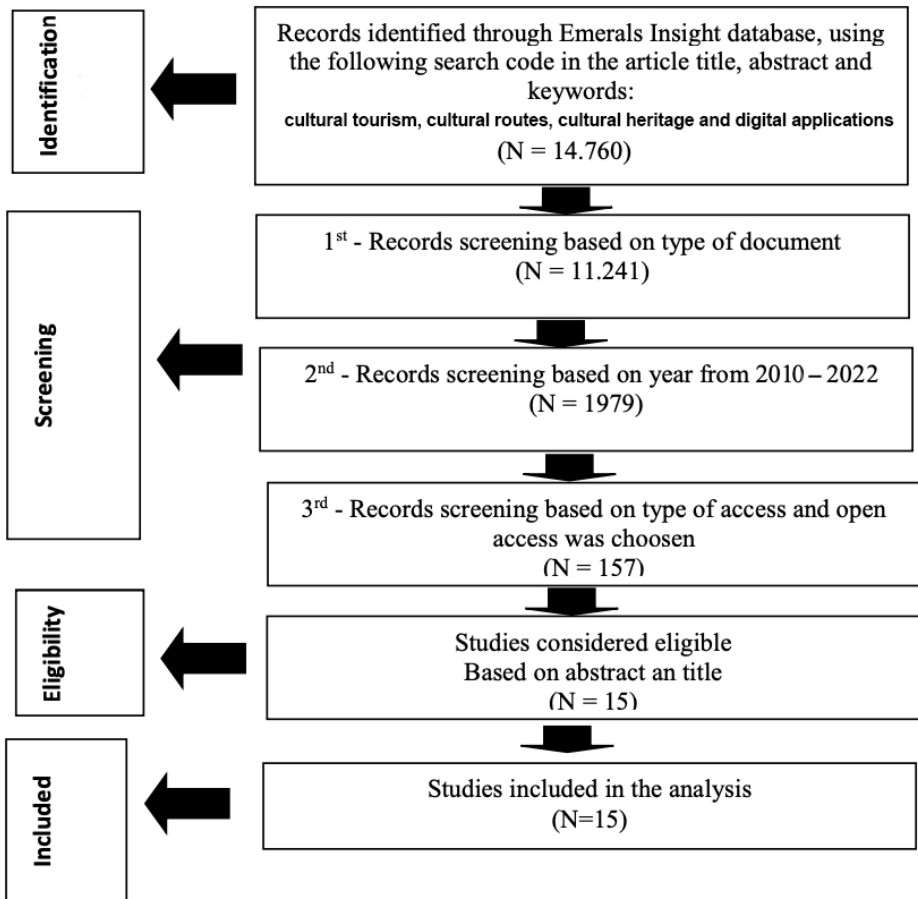

**Figure 1.** PRISMA flow diagram of the article selection process.

The articles chosen in the preceding phase underwent analysis through a two-step process. Initially, a descriptive examination was conducted on the following aspects: the distribution based on the journal, scientific area, and authorship, along with the distribution

by the context of sustainability in tourism where the study was conducted. Subsequently, in the second stage, where both empirical and theoretical studies were identified, a content analysis of the empirical papers was undertaken.

This analysis aimed to identify the following elements: research methods (including data collection, type of data, and sources), methods of analysis, and the results obtained in relation to cultural routes in the tourism sector.

### 3.2. Digital Applications

In this section, we will examine the most significant and widespread of these efforts, emphasizing both their potential in terms of depicting and creating cultural tours and their instances of implementation by cultural organizations and institutions worldwide.

First, a platform for managing cultural data is called PLUGGY [28] (Pluggable Social Platform for Heritage Awareness and Participation). The purpose of the PLUGGY platform is to highlight local and national cultural heritage and to interconnect cultural data, offering users communication and interaction services on cultural topics. Through PLUGGY, users can share their cultural experiences and knowledge about their local culture, essentially contributing to the promotion of Europe's cultural heritage. In this way, PLUGGY enables European citizens, even in less developed regions, to actively participate in activities related to local cultural heritage while facilitating their connection to the production of cultural content at both European and international levels. Through collaboration and interaction, users of the PLUGGY social networking platform will be able to develop a shared cultural and intellectual heritage and contribute to shaping a comprehensive European cultural landscape. Many applications have already been developed that showcase the content of this platform through various approaches. For example, PLUGGY's augmented reality application includes virtual models and overlaid information suitable for museums, both indoors and outdoors. Additional applications inform the user about points of interest around them based on their current location, while others include audio tours that enrich the cultural experience.

The platform Historypin [29,30] is another online platform for cultural content that helps collect memories related to locations through the digitization of old photos. By combining modern images with digitized old photographs, the application creates a collection of memories associated with locations, "transporting" the user to the past by reviving personal memories. With the help of GPS technology for content location, users can navigate to any point on the map. The results can be filtered by date, ranging from 1840 (the period when the oldest photos are dated) to the present day. The application uses an augmented reality camera that "places" historical images from the database onto the landscape being filmed in real time. Furthermore, the platform provides social networking features for users, such as commenting on "uploaded" photos and sharing their own experiences at historical locations, all of which are displayed on a shared cultural map for public access.

The Wikiloc—World Routes [31] platform offers users nature routes (cycling, hiking, sailing, etc.) from around the world. Visitors to the platform can view these routes on the map, categorize them based on difficulty, duration, and distance, read comments and ratings from other users, and see characteristic photos of the route. Additionally, they can suggest and create their own routes and share them with other platform users.

The "Theatres of Epirus" is a cultural route that promises many journeys together in space, time, and the world of the five senses. Five archaeological sites and their theaters are the central stations, the starting points for each person to collect and compose the experiences that inspire and interest them. Everything is here: charming monuments and archaeological sites with exceptionally interesting history, architecture, and craftsmanship, as well as delicious samples of an ancient, unique tradition, the incredible nature of Epirus. The "Theatres of Epirus" are part of the proposal of the "Diadromi" and aim to organize, in a first pilot stage, a cultural route that shares ancient venues for viewing and listening in Epirus. The goal is for this route to serve as a model for others in different regions of Greece [32].

Clio Muse is an interactive guided tour application that shares unique and interesting stories about selected exhibits with a visible time duration. Cultural institutions can process the statistics resulting from user interaction with the exhibits, evaluate them, and plan additional actions accordingly. Curators can manage the content within the application on their own. Finally, they can organize additional activities and engage their visitors by publishing new stories based on their interaction with the application [28].

Another case of virtual touring in a cultural site is the virtual "guide" in Petra, Jordan. As part of Project Zamani [33], modern techniques of digital monument scanning (3D scanning) were used to create three-dimensional models that complement and "explain" the perception of the site to the visitor. These models were integrated into the digital touring application, creating an interactive virtual landscape with panoramic views of objects and buildings, maps, plans, and other spatial content information. The three-dimensional virtual tour of the archaeological site of Petra is a very important tool for visitors who may not be able to physically visit the specific location, as well as for visitors who wish to plan a future visit or "relive" their recent experience.

Also, we need to emphasize the recent research efforts carried out in European initiatives like RURALLURE and RURITAGE [34], which have explored the possibilities of pilgrimage routes as intersections between cultural heritage, tourism, and rural development using digital technologies and narration [35].

In addition, numerous authors have examined the economic ramifications stemming from forms of slow tourism that unfold along geographical routes. They underscore the economic potential of this sector, elucidating both direct and indirect impacts. It has been demonstrated that slow tourism when implemented along a specific route, can foster a more equitable distribution of benefits. Moreover, it can establish a virtuous supply chain that extends not only to the directly traversed territories but also to the surrounding areas. This is attributed to the fact that a significant portion of slow travelers' consumption comprises goods and services that generate additional jobs, both directly and indirectly. A prime illustration of such a supply chain is evident in agricultural production and the processing of food along pilgrimage routes like the Way of Saint James to Santiago de Compostela in Spain [36]. Moscarelli, for instance, calculated the "employment multiplier", indicating the number of jobs generated directly, indirectly, and induced by a one-million-euro increase in visitor demand. The outcome reveals that each euro spent by a slow traveler generates up to 18% more jobs compared to the expenditure of other types of tourists [37]. Consequently, even a relatively small number of pilgrims, in contrast to mainstream tourists, can wield a substantial qualitative impact on territorial balance and rural development [38].

Furthermore, the CHESS project is a research initiative that spans three museums: the Acropolis Museum in Athens, the Stedelijk Museum in Amsterdam, and the Cité de l'Espace in Toulouse. Its main objective is to enhance the museum experience by providing personalized interactive storytelling, rekindling a sense of exploration and wonder among visitors. This framework leverages personalized data to craft individualized narratives that serve as guides during a museum visit. It employs a range of techniques, including mixed reality and game elements, such as digital storytelling and augmented reality on mobile devices [39].

In a similar vein, the EMOTIVE project explores and designs emotionally engaging plot-based narratives that span various genres, such as romance, comedy, and mystery. The project aims to create non-linear narratives that evoke strong emotional responses in people. It distinguishes itself from the CHESS program and other similar projects by focusing on personalization, tailoring stories to the unique needs and characteristics of each visitor. Additionally, the project seeks to provide stable narratives that blend online and on-site experiences, catering to users who wish to access content before, during, or after their visit to a cultural site [40].

Also, the TRACCE project centers on narratives written by travelers during the 18th to early 20th centuries. In this project, the narratives are richly illustrated, bringing to life the environments and daily life that each traveler encountered. The TRACCE project has

created a digital platform that simulates the journey of a traveler and allows users to follow his story through their personal smart devices. A notable characteristic of the TRACCE platform is its capability to customize cultural information according to the user's profile and interests. In order to accomplish this, users are required to fill out a questionnaire that assists in matching each user with predefined profiles. The questionnaire takes into account the user's current location and interests. Users can then choose their preferred route, and the platform guides them, starting from the nearest point of interest relative to their exact location [41].

Finally, TRIPMENTOR had the objective of creating a bilingual (Greek and English) mobile tourist guide for the Attica region. This guide was founded on a well-researched tourist typology, incorporating personalized routes and incorporating gamified elements to offer visitors a distinct and memorable experience. The primary goals of the project were to comprehend and address the potential needs of users, gaining insight into their mindsets, to enable users to immerse themselves in the local culture while bridging modern and ancient Greek culture, drawing from both prejudices and stereotypes, and to stimulate interaction and exchange, encouraging visitors to engage with present-day Attica [13,42].

### 3.3. Digital Application Analysis

As mentioned before, the primary objective of cultural routes is two-fold. First, it aims to enhance the visibility, promotion, and interpretation of cultural heritage through the utilization of new media and technologies. Second, it seeks to augment the user's tour by providing additional (digital) information, presenting it as a cohesive narrative. Furthermore, the goal is to emphasize shared cultural elements across diverse locations and cultural spaces, fostering synergies among cultural entities. The overarching aim includes promoting intercultural dialogue and contributing to sustainable development. Hence, the previously mentioned digital applications can be examined in terms of their features and characteristics, as outlined in Table 1 below.

**Table 1.** Comparison between digital applications in cultural routes.

| App Name | Create Route | Technologies | Mobile | User Location and Map | Social and Sharing |
|---|---|---|---|---|---|
| Pluggy | Yes | AR, VR, DS, 3D | Yes | Yes | Yes |
| Historypin | Yes | AR, DS | Yes | Yes | Yes |
| Wikiloc | Yes | DS | No | Yes | Yes |
| Theatres of Epirus | Yes | DS | No | Yes | Yes |
| Clio Muse | Yes | AR, VR, DS, 3D | Yes | Yes | Yes |
| Zamani project | Yes | AR, 3D | Yes | Yes | Yes |
| RURALLURE | No | DS | No | Yes | Yes |
| RURITAGE | No | DS | No | Yes | Yes |
| CHESS | Yes | AR, VR, DS, 3D | Yes | Yes | Yes |
| Emotive | Yes | VR, DS, 3D | Yes | Yes | Yes |
| TRACCE | Yes | AR, VR, DS, 3D | Yes | Yes | Yes |
| TRIPMENTOR | Yes | AR, VR, DS, 3D | Yes | Yes | Yes |
| Google Art Project | Yes | VR, DS | No | Yes | Yes |
| Echoes | Yes | AR, VR, DS, 3D | No | Yes | Yes |

Augmented Reality: AR, Virtual Reality: VR, Digital Storytelling: DS, 3D models: 3D.

## 4. User Needs and Expectations in Cultural Routes

Based on previous research regarding digital applications in cultural routes, another important aspect is to evaluate and examine user needs and expectations during a cultural

route. Our research was conducted using data analysis methods derived from secondary research, interviews, and questionnaires, which were subsequently analyzed with the assistance of statistical methods. More specifically, a questionnaire was shared via Google Drive (Google Form) for end-users of a cultural route.

The questionnaire that was developed aims to extract information regarding:

- Visitor profiles and traveler backgrounds.
- Motivation for cultural route and specific preferences.
- Level of satisfaction from the cultural route.
- The process and means of cultural routes.

The user questionnaire included demographic questions (age, gender, education, etc.) and questions related to the user's experience during the cultural route (how often they travel, means of transportation, the purpose of the trip, available time, past experiences with cultural routes, digital tools used or desired for the route, etc.). Based on the collection of 89 questionnaires and the results obtained for the users, we present, indicatively, the graphical representation regarding some of the findings in Figures 2 and 3 below.

Studying the user questionnaire, we easily observe that the educational level of the participants is high, with 82% stating they come from tertiary education. This is confirmed by the age distribution, where 73% fall between the ages of 26–50, an age group in which most have completed their studies. Additionally, approximately 74% are employed, and around 15% are students. This partially justifies the high percentage of good physical condition (76.4%) and a high level of familiarity with new technologies (79.3% answered 4 and 5 on a scale of 1 to 5).

Additionally, it is apparent that most individuals set out to travel with family or friends up to three times per year, utilizing cars or buses as their primary mode of transportation. Furthermore, a considerable proportion of vacationers prefer walking, which presents an opportunity to strategize activities and applications that will enrich the cultural and tourist experience. This is particularly noteworthy given that the trips' primary objective is to visit tourist attractions, with urban exploration ranking third and culture the second most desired destination.

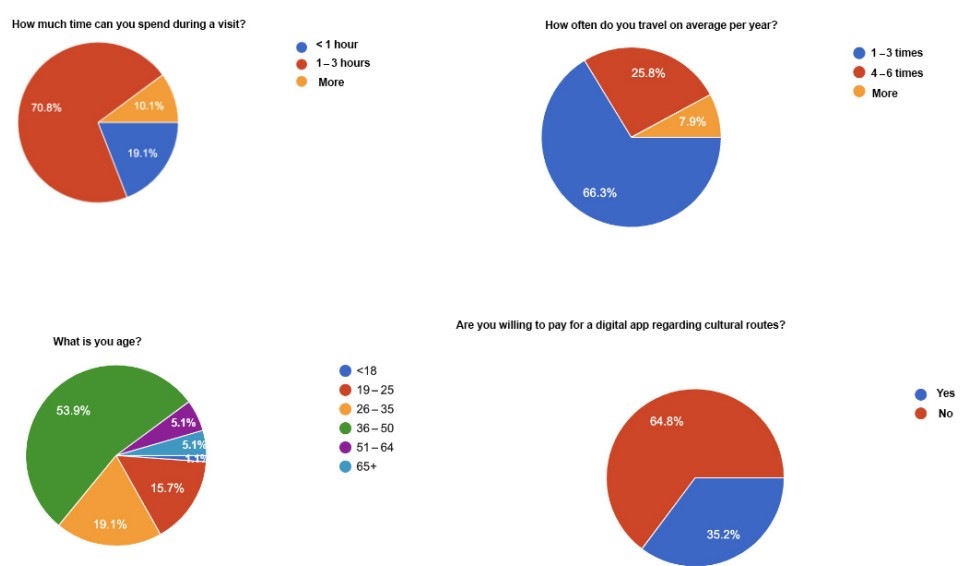

**Figure 2.** Evaluation results and user analysis.

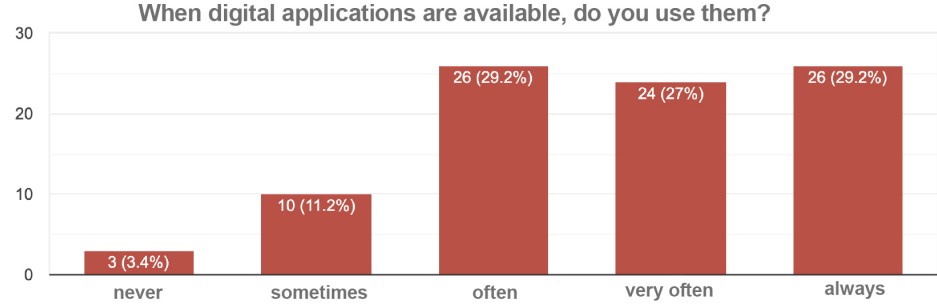

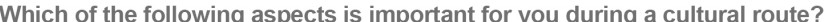

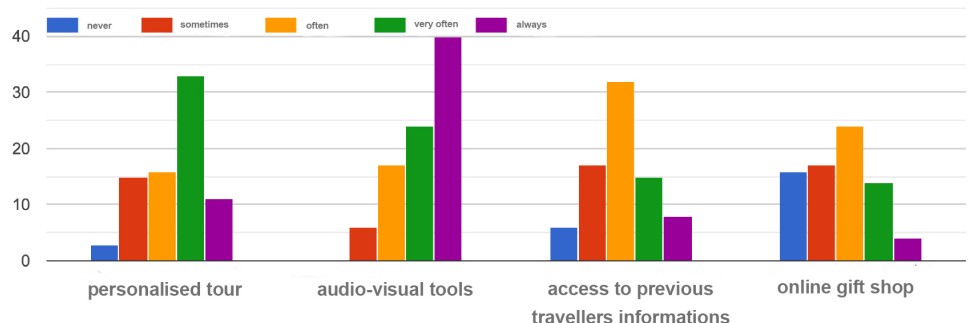

**Figure 3.** Evaluation results—digital apps.

The website, parking facilities, and suitable infrastructure for people with disabilities are the most crucial elements for modern travelers. The observation that visitors are very familiar with new technologies largely explains their desire for interactive systems and applications, as well as their willingness to delve deeper into what they have visited. In this context, it is not surprising that almost everyone gets information about their trips or visits from the Internet and desires the existence of digital applications and enriched experiences, even though the majority chooses traditional cultural spaces such as archaeological sites and art galleries.

However, the majority (64.8%) would not pay for access to digital applications and assistants, although it is equally true that the percentage of those who would pay is quite high (35.2%). This can be interpreted by the lack of familiarity with the cultural tourism product market and the economic crisis, but it certainly leaves room for further development in line with parallel advancements in the field of information technology and digital products.

## 5. Discussion and Conclusions

### 5.1. Discussion

The purpose of this review is to ultimately address one of the demands of today's conditions in our country, focusing on forms of tourism that cater to specific motivations. It thus concentrates on one of these forms of tourism that show particular dynamism and prospects for revitalizing our national tourism product, and that is cultural tourism. The modern tourist, as revealed by the results of our research, presents a variety of motivations. Therefore, destinations must necessarily offer complex tourism experiences enriched with alternative activities that differentiate from the classic model of mass tourism.

*5.2. Theoretical Implications*

Visitors, as a whole, seek both entertainment and a learning (or broader educational) experience. This requirement necessitates enhancing the interpretative perspective of cultural applications in tourism at the core of the planned public and private interventions. Additionally, cultural tourism is perceived in the broader sense of the tourism industry, including the methods and technologies serving the acceptance, understanding, experience, diffusion, protection, and preservation of natural and cultural resources, as well as the conduct of activities that adhere to international tourism standards. Therefore, a primary goal of cultural tourism is to enable every traveler to become a kind of cultural anthropologist.

Finally, the general components of cultural tourism can be defined both based on tourism conducted for cultural purposes and as a high-quality service technology conforming to international tourism standards and the methodology of presenting natural and cultural heritage to the public. The primary research conducted, in combination with the literature review, has shown that the cultural and natural wealth of a region rich in cultural heritage can serve as a comparative advantage for modifying the tourism product and developing cultural tourism using cultural routes as a tool.

*5.3. Practical Implications*

The use of tools and applications for highlighting cultural heritage is expected to attract visitors with special interests, extend the tourist season in each respective area, and generate economic, social, and cultural benefits by leveraging human potential and improving the quality of life in local communities. Simultaneously, the incorporation of cultural assets into contemporary social life through the implementation and operation of cultural routes will protect and promote cultural heritage. Therefore, the encouragement of partnerships between cultural and tourism stakeholders is expected to boost growth and contribute to the fundamental principle of sustainability: the reinvestment of development.

However, to carry out this endeavor, the difficulties and problems that may arise must be overcome, such as the absence of a strategic tourism development plan by several local authorities, frequent lack of interest from the local community in the utilization and protection of cultural heritage, coupled with a frequent lack of collaboration between the local community and relevant archaeological services.

In conclusion, the application of these technologies in infrastructure development proves to be one of the main goals and characteristics of cultural tourism. As such, we can proceed with the design of tours that promote culture and the value of monuments, contributing to an infrastructure of supply that caters to different types of demand. This, in turn, will support the fundamental principle of sustainability: development reinvestment.

## 6. Challenges and Limitations

Designing successful cultural routes, especially those that highlight the literary tradition and heritage of a region, depends on multiple parameters. As emerges from the relevant literature, various issues need to be resolved. One significant problem faced by several modern applications, particularly critical in the case of a cultural-touristic application, is the lack of intercultural design.

A digital guiding application is inevitably used by users from diverse cultural backgrounds to a degree where the application's interface may appear unwieldy or even unpleasant to specific user groups. Therefore, beyond the necessary choice of an appropriate user interface language and the available information and media provided by the application, the general design philosophy that permeates the entire functionality of the application should adapt to the requirements of the respective cultural groups. Other considerations that must be taken into account include the digitization of available information and sources, particularly in a format suitable for further processing and presentation using new technologies.

Moreover, the well-known "cold start problem" meaning the inability of a system to generate effective output data due to insufficient input data, is a critical parameter in

personalized information delivery functions [43]. This difficulty hinders the immediate and effective recognition of user interests and profiles. As a result, similar functions may fail initially or request much information from the user at the beginning of the digital journey, which may not always be pleasant for visitors eager to start their exploration. Additionally, although today, the average visitor to a destination is technologically savvy, there may be user groups that lack the appropriate technological training. Therefore, providing indicative training or offering assistance and explanations during application use are generally essential and helpful tools. Finally, obstacles to the smooth operation of a digital cultural route and, consequently, to the overall user experience could also be technical issues related to the user's device. For example, a lengthy route with rich multimedia content may place significant demands on battery consumption or device connectivity.

Also, recognizing the multidisciplinary nature of cultural route design, we must consider in the future the necessity of collaborating with experts from diverse fields, including cultural studies, human-computer interaction, and tourism management. This interdisciplinary approach has the potential to enhance the discussion and offer a more comprehensive perspective on the challenges.

In conclusion, designing cultural routes and applications that cater to diverse cultural backgrounds involves addressing issues related to intercultural design, adapting the design philosophy to different cultural groups, digitizing information appropriately, considering the "cold start problem", and providing technical support for users with various levels of technological proficiency. These considerations are essential for creating a successful and user-friendly cultural experience.

**Author Contributions:** Conceptualization, M.K. and A.T.; methodology, E.I.; software, E.I.; validation, E.I., A.T. and M.K.; formal analysis, E.I.; investigation, E.I.; resources, E.I. and G.K.; data curation, E.I.; writing—original draft preparation, E.I., G.K. and M.K.; writing—review and editing, E.I., A.T. and M.K.; visualization, E.I.; supervision, M.K.; project administration, G.K. and M.K.; funding acquisition, M.K. All authors have read and agreed to the published version of the manuscript.

**Funding:** This research received no external funding.

**Institutional Review Board Statement:** Not applicable.

**Informed Consent Statement:** Not applicable.

**Data Availability Statement:** Not applicable.

**Conflicts of Interest:** The authors declare no conflict of interest.

## Abbreviations

The following abbreviations are used in this manuscript:

| | |
|---|---|
| CR | Cultural Routes |
| CH | Cultural Heritage |
| HM | Heritage Management |
| CT | Cultural Tourism |
| CUX | Cultural User eXperience |

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
