# Peer review of "Analyzing Cultural Routes and Their Role in Advancing Cultural Heritage Management within Tourism: A Systematic Review with a Focus on the Integration of Digital Technologies"

_encyclopedia, doi:10.3390/encyclopedia3040108_

Round 1

Reviewer 1 Report

Comments and Suggestions for Authors

It is a good review essay on an important topic. As such it involves collecting materials on the topic and presenting them. It does this well. It is a useful essay that is appropriate for it publication outlet. I would publish.

Author Response

First of all, we would like to thank you for your effort in revising our manuscript. We really appreciate the careful review and constructive suggestions. In what follows, we try to address all the points raised in the review. The manuscript is now substantially improved after making the suggested edits, answered seperately in the attached document.

Reviewer 2 Report

Comments and Suggestions for Authors

The study is devoted to the use of digital technologies in cultural tourism. However, the title of the article does not reflect its content.

The current version of the article title, “Analyzing Cultural Routes and their Role in Advancing Cultural Heritage Management within Tourism: A Literature Review,” suggests a scientometric review. Writing such structured reviews requires a certain methodology and structure of the article. For example, I would advise authors to consider the PRISMA or SPAR-4-SLR approaches for preparing a scientometric review study. In the current version of the text, the article is not a research article. The abstract, title and keywords do not fully reflect its content.

There is no clearly stated purpose of the study at the beginning of the article. Also in the introduction, the authors need to clearly justify the relevance of this study based on the scientific literature.

Authors need to add a methodological section and describe the methodology and algorithm for conducting the Literature Review. I would also advise the authors to focus specifically on the problem of using digital technologies in cultural tourism.

In section “3. Cultural Routes: State of the Art" essentially provides a description of digital services and platforms for tourists. This section requires reorganization and the addition of an analytical component, for example, comparing services with each other and justifying a closer connection between digitalization and the development of cultural routes of various types.

In their conclusions, authors should reflect the value and scientific novelty of their research. In its current form this is not obvious.

Authors must prepare a list of references according to the journal's guidelines (for example, a number of sources do not indicate the year). And also expand the list of sources used by including recent studies on the topic of research.

I did suggest that the authors take a systematic review approach using PRISMA or SPAR-4-SLR techniques. This recommendation of mine is intended to enhance the value of the study and is not in the nature of a requirement.

However, in my opinion, the article has a significant drawback. In its current format, the article lacks a clear logic and coherence between sections, as well as between the main text of the article and conclusions.  

My recommendation is due to the fact that the authors are trying to present a comprehensive analysis of the scientific literature on the problem of developing cultural tourist routes, touching on many aspects at once:  

Definition of cultural tourism and cultural route,  

Cultural experience of tourists

Review of digital applications and services

Management of cultural routes.  

However, the description of these aspects is quite superficial. This is also evident from the presented list of references. Almost half of the sources are dated before 2010. Modern research in recent years (2021-2023) is practically not presented. Meanwhile, the authors raise the issue of using digital technologies in obtaining new user experiences and the problem of developing successful cultural routes at the present stage.  

A striking example is the section “3. Cultural Routes: State of the Art", where authors provide a coherent description of digital services, but do not provide any critical analysis. Obviously, the above list is not exhaustive; the criteria for its compilation and the purpose of the description are not clear. A critical analysis is needed. If the article format is left as a literature review, then the text requires significant revision to increase its logic and coherence, as well as focus on a specific problem. I suggest that authors narrow the topic but provide a more in-depth review of the literature, incl. recent research on the topic.

Author Response

First of all, we would like to thank you for your effort in revising our manuscript. We really appreciate the careful review and constructive suggestions. In what follows, we try to address all the points raised in the review. The manuscript is now substantially improved after making the suggested edits, answered separately in the attached document.

Reviewer 3 Report

Comments and Suggestions for Authors

The paper provides an interesting review within the field of cultural tourism, with a particular focus on the concept of cultural routes. It is an interesting read that many researchers may find useful.

The only pitfall is that the review seems to miss important developments and theoretical references from the last 2-3 years. I am providing some pointers and references below that I believe the authors should include in their analysis.

To begin with, many authors have analysed the economic effects generated by the forms of slow tourism that develop along territorial lines, highlighting the economic potential of this sector both for the direct, indirect and induced impacts. It has been shown that slow tourism activated along a line can encourage a more equitable and balanced dissemination of the benefits and that a virtuous supply chain can be established, with which to involve not only the territories directly crossed, but also the surrounding ones, due to the fact that most of the slow travellers’ consumption is made up of goods and services that generate more jobs (direct and indirect). One of the clearest examples of a supply chain is certainly that linked to agricultural production and the transformation of food along pilgrimage routes like the Way of Saint James to Santiago de Compostela in Spain. Moscarelli, for example, calculated the so-called “employment multiplier”, i.e. the number of jobs generated directly, indirectly and induced, with an increase of one million euros in the visitor’s final demand. The result is that each euro spent by a slow traveller generates up to 18% more jobs than the expenditure of another type of tourist. Thus, even a quantitatively modest number of pilgrims –in comparison with mainstream tourists– can have a qualitatively high impact in territorial balance and rural development.

I would also like to highlight the work conducted recently in European projects such as RURALLURE and RURITAGE, which has investigated the potential of pilgrimage routes at the crossroads between cultural heritage, tourism and rural development. The most recent references below come from RURALLURE, touching different aspects relevant to the review: technology, narratives, etc.

Recommended references:

Kádárné Kelemen, I., Šramová, B., Deptová, T., & Vas R. (2023). The rurAllure network contributing to the renaissance of pilgrimage culture in Europe – A case of the Way of Mary. Euro-Asia Tourism Studies journal. https://doi.org/10.58345/JBQN5784

Gomez-Heras, M.; González Soutelo, S.; Castelo Ruano, R.; García Juan, L. The Challenge of Accessibility to Heritage around the Via Francigena: The Potential of Thermal Heritage for Accessible Tourism. Heritage 2023, 6, 7115-7125. https://doi.org/10.3390/heritage6110371

Vranić, V., Lang, J., Nores, M.L. et al. Use case modeling in a research setting of developing an innovative pilgrimage support system. Univ Access Inf Soc (2023). https://doi.org/10.1007/s10209-023-01047-1

Bassani, M., Bergamo, M., & González Soutelo, S. (eds.) (2023) Archaeology & Pilgrimage. “La Rivista di Engramma” n.204. https://www.engramma.it/eOS/

López Nores, M., Pazos Arias, J.J., Reboreda Morillo, S., & Penín Romero, Ó. (2023) “La Rivista di Engramma” n.204, luglio/agosto 2023, pp. 11-20. doi: https://doi.org/10.25432/1826-901X/2023.204.0002

Lois González, R. C., & López, L. (2021). The singularity of the Camino de Santiago as a contemporary tourism Case. In P. Pileri & R. Moscarelli (Eds.), Cycling & walking for regional development. How slowness regenerates marginal areas (pp. 221–234). Cham, Germany: Springer.

Pileri, P., & Moscarelli, R. (Eds.) (2021) Cycling & walking for regional development. How slowness

regenerates marginal areas. Cham, Germany: Springer.

Moscarelli, R. (2021) Slow tourism, public funding and economic development. A critical review on the case of the Way of St. James in Galicia. Revista Galega de Economía, 30 (3), 7522. DOI: 10.15304/rge.30.3.7522

Fernández-Fernández, M., Lazovski, O., & Real Neri, G. (2020) Tourism impacts in an emerging destiny through the local entrepreneurship perception: The Fisterra case. Journal of Tourism and Heritage Research, 3(2), 269-285. Retrieved from http://www.jthr.es/index.php/journal/article/view/175

Med Pearls project (2021) Research study on Slow Tourism international trends and innovations. http://www.slow-tourism.net

Moira, P., Mylonopoulos, D., & Kondoudaki, A. (2017) The application of slow movement to tourism: Is slow tourism a new paradigm? Journal of Tourism and Leisure Studies, 2(2). DOI: 10.18848/2470-9336/CGP/v02i02/1-10

Allgemeiner Deutscher Fahrrad-Club (2019) Travelbike Bicycle Travel Analysis – Summary report. Berlin, Germany.

Downward, P., Lumsdon, L., & Weston, R. (2009). Visitor expenditure: The case of cycle recreation and tourism. Journal of Sport and Tourism, 14(1), 25-42. DOI: https://doi.org/10.1080/14775080902847397

Dunkelbergt, D., & Püschel, R. (Eds.) (2009) Grundlagenuntersuchung Fahrradtourismus in Deutschland. Berlin, Germany: Bundesministerium für Wirtscha und Technologie. Retrieved from https://digital.zlb.de/viewer/resolver?urn=urn:nbn:de:kobv:109-opus-259062

Piket, P., Eijgelaar, E., & Peeters, P. (2013) European cycle tourism: A tool for sustainable regional rural development. Applied Studies in Agribusiness and Commerce, 7(2-3), 115-119. DOI: 10.19041/Apstract/2013/2-3/19

Pileri, P., Giacomel, A., & Giudici, D. (2015) VENTO. La rivoluzione leggera a colpi di pedali e paesaggio. Mantova, Italia: Corraini. reach out!

Author Response

(The authors gave the same response as above.)

Reviewer 4 Report

Comments and Suggestions for Authors

Dear Authors,

I appreciate the effort put into your manuscript on designing cultural routes, and I believe your work has the potential to make a valuable contribution to the field. However, the manuscript could benefit from refinement to enhance the overall clarity and impact of your research:

Abstract: Condense the abstract for conciseness and clearly articulate the main objective and contributions. Reorganize the content logically, starting with the motivation and then methodology and key findings. Additionally, highlight the novel aspects or contributions of the paper.

Introduction: Improving the transition between different aspects of the paper is crucial for sustaining reader engagement. For example, when shifting from the discussion on cultural routes to the literature review on technological challenges (from sentences 41 to 42), a more seamless connection is needed. While the introduction notes a "comprehensive literature review within the field of cultural tourism," specifying the scope and boundaries of this review is essential. Clearly outlining which specific aspects of cultural tourism will be covered and establishing the criteria for inclusion would provide readers with a clearer understanding. Creating a more cohesive narrative requires better linkage between sentences, especially between the discussion on technological challenges and the exploration of cutting-edge technologies in managing cultural heritage (from sentences 42 to 43). Additionally, the contribution of the study should be included in this section.

Literature Review: Elaborate on the concept of "Points of Interest (POI)" in paragraph 106 by providing examples and explaining their contribution to the broader idea of a cultural route. Strengthen the narrative by emphasizing the connection between cultural routes and tourism, highlighting how cultural routes enhance the overall cultural tourism experience. Clearly articulate the link between Cultural User Experience and cultural routes, explaining how digital applications, like Svevo Tour, Archeoguide, and MuseumEye, contribute to and enhance the user experience within the context of cultural routes.

Planning of a Cultural Route: The authors should clarify the significance of López Fernández's facets in planning cultural routes, elucidating their specific contributions for a stronger argument. Additionally, incorporating effective transition phrases between the discussion of objectives and López Fernández's facets would improve the logical flow of ideas. To enhance the section on accessibility, expanding on why it is a crucial factor in cultural routes and delving into how it contributes to an enriched tourism experience would strengthen the content.

Discussion/Conclusion: To enhance the discussion and conclusion section, the authors should improve clarity and organization by introducing subheadings like discussion, theoretical implications, and practical implications. They should begin with a concise summary of the main findings, reinforcing the connection to the research objectives and addressing the current demands in the country. The authors should integrate specific research results into the discussion, particularly when mentioning the modern tourist, providing examples to illustrate diverse motivations. Additionally, connect the current findings with existing literature to provide a broader context and highlight the novelty or contribution of the present study.

Challenges and Limitations: Given the multidisciplinary nature of cultural route design, the authors could acknowledge the need for collaboration with experts from fields such as cultural studies, human-computer interaction, and tourism management. This interdisciplinary approach could enrich the discussion and provide a more holistic view of the challenges.

Author Response

(The authors gave the same response as above.)

Round 2

Reviewer 3 Report

Comments and Suggestions for Authors

The authors have addressed my comments and those from the other reviewers satisfactorily, and the article has improved significantly in comparison with the first version.

Reviewer 4 Report

Comments and Suggestions for Authors

I am pleased to confirm that all the raised points have been thoroughly addressed. The modifications have significantly strengthened the coherence and clarity of the content, enhancing the overall quality of the paper. I am impressed with the careful consideration given to each suggestion, resulting in a more robust and compelling narrative. Therefore, I wholeheartedly recommend the acceptance of the manuscript for publication